# A Mouse Model of Mild *Clostridioides difficile* Infection for the Characterization of Natural Immune Responses

**DOI:** 10.3390/microorganisms12101933

**Published:** 2024-09-24

**Authors:** Assaf Mizrahi, Gauthier Péan de Ponfilly, Diane Sapa, Antonia Suau, Irène Mangin, Aurélie Baliarda, Sandra Hoys, Benoît Pilmis, Sylvie Lambert, Anaïs Brosse, Alban Le Monnier

**Affiliations:** 1Service de Microbiologie Clinique, Hôpitaux Saint-Joseph & Marie-Lannelongue, 75014 Paris, France; gpeandeponfilly@ghpsj.fr (G.P.d.P.); bpilmis@ghpsj.fr (B.P.); alemonnier@ghpsj.fr (A.L.M.); 2Institut Micalis UMR 1319, Université Paris-Saclay, INRAe, AgroParisTech, 91400 Orsay, France; diane.sapa19@gmail.com (D.S.); aurelie.baliarda@inrae.fr (A.B.); sandra.hoys@universite-paris-saclay.fr (S.H.); sylvie.lambert-bordes@u-psud.fr (S.L.); anais.brosse@universite-paris-saclay.fr (A.B.); 3USC ANSES-Cnam Metabiot, Conservatoire National des Arts et Métiers, 75003 Paris, France; antonia.suaupernet@lecnam.net (A.S.); irene.mangin@lecnam.net (I.M.)

**Keywords:** *Clostridioides difficile*, immune response, colonization factor, toxins, gut microbiota

## Abstract

(1) Background: We describe a model of primary mild-*Clostridioides difficile* infection (CDI) in a naïve host, including gut microbiota analysis, weight loss, mortality, length of colonization. This model was used in order to describe the kinetics of humoral (IgG, IgM) and mucosal (IgA) immune responses against toxins (TcdA/TcdB) and surface proteins (SlpA/FliC). (2) Methods: A total of 10^5^ CFU vegetative forms of *C. difficile* 630Δ*erm* were used for challenge by oral administration after dysbiosis, induced by a cocktail of antibiotics. Gut microbiota dysbiosis was confirmed and described by 16S rDNA sequencing. We sacrificed C57Bl/6 mice after different stages of infection (day 6, 2, 7, 14, 21, 28, and 56) to evaluate IgM, IgG against TcdA, TcdB, SlpA, FliC in blood samples, and IgA in the cecal contents collected. (3) Results: In our model, we observed a reproducible gut microbiota dysbiosis, allowing for *C. difficile* digestive colonization. CDI was objectivized by a mean weight loss of 13.1% and associated with a low mortality rate of 15.7% of mice. We observed an increase in IgM anti-toxins as early as D7 after challenge. IgG increased since D21, and IgA anti-toxins were secreted in cecal contents. Unexpectedly, neither anti-SlpA nor anti-FliC IgG or IgA were observed in our model. (4) Conclusions: In our model, we induced a gut microbiota dysbiosis, allowing a mild CDI to spontaneously resolve, with a digestive clearance observed since D14. After this primary CDI, we can study the development of specific immune responses in blood and cecal contents.

## 1. Introduction

*Clostridioides difficile*, a Gram-positive, anaerobic and spore-forming opportunistic bacterium, is the etiological agent of gastrointestinal infections (*Clostridioides difficile* infection CDI) in humans. Clinical presentations range from asymptomatic colonization to severe presentation with potential complications [1]. The use of broad-spectrum antibiotics often leads to dysbiosis in the intestinal microbiota, enabling ingested *C. difficile* spores to germinate, multiply, and colonize the intestinal tract using non-toxin factors such as SlpA and FliC [2,3]. Disease manifestation occurs after the production of major virulence factors, namely TcdA and TcdB toxins, which cause intestinal tissue damage and inflammation [4].

Despite the availability of updated clinical practice guidelines and new treatments for CDI, it remains a significant threat for the healthcare systems worldwide [5].

The development of an immune response to *C. difficile* plays a crucial role in the pathophysiological process and patient outcomes [6,7,8,9,10,11,12,13]. Bezlotoxumab, a monoclonal antibody against TcdB of *C. difficile*, is recommended alongside standard care to decrease the incidence of recurrent CDI [14]. However, little is known about the development of a protective immune response following a primary CDI in individuals with no history of previous exposition to *C. difficile*. Similarly, studies evaluating antibodies against *C. difficile* surface proteins (SlpA, FliC) indicate a protective role for SlpA and FliC IgG, yet these have not been studied extensively post primary CDI [15,16,17].

As humans are frequently colonized during infancy, leading to the production of specific antibodies as early as the first months of life [11,18,19], immunological studies in humans cannot fully address this question and must be performed in animal models. Currently, several animal models have been developed to study various aspects of CDI, such as colonization, virulence, transmission, and recurrence, but only a few for specifically studying immune responses [20].

Among these, the mouse model is the most widely used, due to advancements in methods to induce CDI susceptibility and the availability of mouse-specific reagents for studying the host immune response to infection [21,22]. The severity of CDI in mouse models depends on the antibiotic treatment and the *C. difficile* strain used. For example, animals treated with an antibiotic cocktail and clindamycin before *C. difficile* challenge can exhibit a range of outcomes, from remaining clinically well to 100% mortality 2–4 days after challenge [23]. We adapted a model from Chen et al., using C57BL/6 mice infected with the 630Δ*erm* strain [15,16,24]. This model is intended to be closer to the pathophysiology of human CDI, with mild mortality and spontaneous resolution of clinical signs without the need of anti-CDI antibiotics.

Based on this rationale, our study aims to describe and validate a model of primary mild-*C. difficile* infection (CDI) in a naïve host, focusing on the kinetics of humoral (IgG, IgM) and mucosal (IgA) immune responses against toxins (TcdA/TcdB) and surface proteins (SlpA/FliC). Additionally, we evaluated mucosal IgG and IgA antibody levels towards toxins and surface proteins following a new challenge.

## 2. Materials and Methods

Animal experiments

Six-week-old female C57BL/6 mice (Charles River’s, Saint Germain Nuelles, France) were caged after the verification of the absence of *C. difficile* colonization. All mice used in the experiments were housed in groups of 4 or 5 per cage under the same conditions, as follows: specific pathogen-free conditions, a temperature-controlled environment, and *ad libitum* access to water and diet. Food, water, bedding, and cages were autoclaved. All procedures involving animals were performed according to protocols approved by the Committee on the Ethics of Animal Experiments n° 26 of the University of Paris-Sud and authorized by the French Ministry of Research (APAFIS#23414-2019121910116284).

The protocol to induce dysbiosis was adapted from Chen et al. [23], with the main modification being the use of the 630Δ*erm* exponential cells. A cocktail of several antibiotics was added to the drinking water during the first three days of the experiment (D6 to D3; Figure 1) and was changed daily. Day 6 was selected as the baseline to account for the pre-infection phase, ensuring that the initial immune status and any potential variations were documented before the introduction of *C. difficile.*

The antibiotic cocktail consisted of kanamycin (0.4 mg/mL; Sigma Aldrich, St. Louis, MO, USA), gentamicin (0.035 mg/mL; Panpharma, Luitré, France), colistin (850 U/mL; Sanofi, Paris, France), metronidazole (0.215 mg/mL; Braun, Melsungen, Germany), and vancomycin (0.045 mg/mL; Mylan, Canonsburg, PA, USA). Over the following days of the experiment, the antibiotic cocktail was replaced by water. On day 1, animals received a single dose of clindamycin (10 mg/kg) administrated intraperitoneally. The next day, mice were challenged orally by a gavage of 10^5^ CFU of *C. difficile* 630Δ*erm*.

Vegetative-form inoculum was prepared by an overnight culture of *C. difficile* (toxigenic strain 630Δ*erm* or the non-toxigenic strain ATCC43602 [25]) in Brain Heart Infusion (BHI) medium, and diluted in sterile physiological serum to the concentration required for reproducible colonization.

A total of 48 mice were sacrificed according to six groups at endpoint times on days 6, 2, 7, 14, 21 and 28 of the challenge (Groups A to F). Each animal was directly sampled under anesthesia by intracardiac puncture for blood collection at each time point (Figure 1). After infection, mice were monitored twice a day and weighed daily to monitor weight loss relative to the initial weight measured on D0. Weight loss was monitored from the first day after challenge until return to initial weight. The clinical signs of CDI were observed daily for diarrhea and decreased activity.

Groups G and H consisted of 15 mice initially infected with the *C. difficile* 630Δ*erm* strain and the non-toxigenic strain ATCC43602, respectively, and then challenged again with the same strain at D28 following the same protocol, including the antibiotic cocktail. For both groups, intracardiac blood and cecal content sampling were performed at the end of the assay (D56).

2.Biological sampling

a.Blood and cecal contents sampling for immune analysis

All blood samples were processed in the same manner, as follows: After one hour at room temperature, samples were incubated at 4 °C for three hours and centrifuged at 3000× *g* for 10 min. After settling, sera were collected and stored at −80 °C. For IgA detection, cecal contents were collected at the endpoint, and a cold PBS solution containing the protease inhibitor cocktail (Sigma Aldrich, St. Louis, MO, USA) was used. The suspensions were vortexed for one min and centrifuged at 3000 rpm for 10 min at 4 °C. The supernatant was retained and re-centrifuged in a new Eppendorf at 12,000 rpm for 10 min at 4 °C. The final supernatant was collected and stored at −80 °C.

b.Stool sampling for *C. difficile* colonization rate

To assess the colonization rate, fecal pellets were collected before antibiotic administration (D-6) and on D2, D7, D14, D21, and D28 after orogastric challenge for groups A to F. Colonization was also assessed on the cecal contents at each endpoint time.

Ten mg of feces/cecal contents were suspended in one mL of liquid casein yeast extract (LCY) broth, and 100 µL of tenfold serial dilutions were cultured on Columbia agar containing 5% horse blood, 25% (*w*/*v*) D-cycloserine, and 0.8% (*w*/*v*) cefoxitin. Typical fluorescent colonies were counted under UV light (312 nm).

On days 7 and 14, an enrichment step was performed to enhance the detection of *C. difficile* spores, particularly in cases of low bacterial inoculum. This involved incubating fecal pellets in Brain Heart Infusion (BHI) broth containing 0.1% taurocholate, followed by subculturing on Columbia agar plates. On D2 post challenge (D2 and D30)*, C. difficile*-free toxins were detected in fecal samples diluted in an anaerobic PBS at 10 mg/mL using the TECHLAB^®^ C. DIFF QUIK CHEK COMPLETE^®^according to the manufacturer’s instructions. The presence of *C. difficile*, combined with the detection of free toxins in stool, was used as criteria to objectify CDI and explain the clinical signs.

c.Stool sampling for microbiota analysis

Stool samples were collected from four mice before antibiotic treatment (D6), after three days of exposition to antibiotics (D3), just before the clindamycin administration (D1), and the day of challenge by *C. difficile* (D0) (Figure 1). Samples were weighed and stored at −80 °C in 0.2 mL sterile tubes until the genomic DNA extraction step.

Total DNA was extracted from approximately 125 mg of frozen stool samples using a bead-beating method with 0.1 mm glass beads and a Bead Beater (Biospec Products Inc., Bartlesville, OK, USA), as previously described [26]. Finally, dry pellets were suspended in 100 µL of sterile water and stored at −20 °C.

The bacterial V3–V4 region of the 16S ribosomal gene was amplified from the purified genomic DNA using the AmpliTaq Gold 360 (Applied Biosystems, Waltham, MA, USA) and the primers F343-2 and R784-2. Sequencing was performed using Illumina MiSeq technology on the GeT-PlaGe technical platform (Toulouse, France). Sequences were then analyzed and normalized using the FROGS (Find Rapidly Operational Taxonomic Unit (OTU) with Galaxy Solution) pipeline [27]. PCR primers were removed and sequences with sequencing errors in the primers were excluded. Reads were clustered into ASVs using the Swarm clustering method. Chimeras were removed, and 246 clusters were assigned at different taxonomic levels (from phylum to species) using a RDP classifier and NCBI Blast+ on SILVA_138_16S_pintail 100database [28]. The raw 16S rRNA sequencing data supporting the findings of this study have been deposited in the NCBI Sequence Read Archive (SRA) under the accession number PRJNA1077911. The phyloseq R package and associated tools in the R environment were used to carry out analysis, including the alpha and beta diversity analyses [29,30]. Kruskal–Wallis tests were used for statistical analysis, followed by Wilcoxon tests when appropriate. Beta diversity statistics were calculated using pairwise PERMANOVA with 999 permutations (vegan R package) [31]. It was used to build principal coordinates analysis (PCoA) plots (ggplot2 and plotly R packages). The linear discriminant analysis (LDA) effect size (LEfSe) algorithm was used to identify taxa that were specific to each time point using Galaxy solution [32].

3.Antigen preparation

a.Toxins

As previously described, TcdA and TcdB were native toxins produced from *C. difficile* strain R20291 and purified by double chromatography (gel filtration and ion exchange) [33].

b.Non-toxin antigens

As previously described by our team, SlpA and FliC were recombinant proteins issued from the 630Δ*erm* strain, produced in *E. coli*, and purified by a single-step affinity chromatography (BD Talon cobalt affinity resin, BD Biosciences, San Jose, CA, USA) [15,16].

4.Detection of specific antibodies by ELISA

Systemic anti-toxins (TcdA and TcdB), anti-SlpA, anti-FliC IgG, and IgM levels were measured by ELISA as previously described [15]. Mucosal IgA antibody levels were determined on cecal samples. Briefly, coating was performed using carbonate/bicarbonate buffer (pH 9.5); the toxin concentration was 1 µg/mL and non-toxin antigen concentration was 5 µg/mL. Sera were diluted 1/20 for IgG and 1/10 for IgM. Cecal contents were not diluted for IgA detection. Detection was performed with a streptavidin–biotin combination. All samples were tested in duplicate and treated simultaneously to avoid inter-assay variation. Assays with antigen in the absence of sera served as negative controls. Immunoglobulin levels were expressed in O.D. (optical density) units obtained at 405 nm.

Serum IgM against TcdA, Tcd,B SlpA, and FliC were tested on D6, D2, D7, D14, and D21. Serum IgG against TcdA, Tcd,B, SlpA, and FliC were tested on D6, D2, D7, D14, D21, D28, and D56. IgA against TcdA, TcdB, SlpA, and FliC were tested on cecal content on D56.

5.Statistical analysis

Descriptive statistics are based on means (+/− standard deviation) or medians [minimum-maximum], depending on the distribution of quantitative variables. Qualitative variables are described as numbers and percentages. Univariate comparisons were made using standard statistical tests after verifying the distribution of the variables. To account for the increased risk of Type I errors due to multiple comparisons, we applied the Bonferroni correction to adjust the *p*-values.

IgM Comparisons: IgM levels were compared across three time points (D6 vs. D7, D6 vs. D14, and D6 vs. D21) for four different antigens, resulting in a total of 12 independent comparisons (4 antigens × 3 time points). The Bonferroni-adjusted significance threshold was calculated by dividing the standard alpha level of 0.05 by the number of comparisons (12), yielding an adjusted *p*-value threshold of 0.00417.IgG Comparisons: IgG levels were compared across four time points (D6 vs. D7, D6 vs. D14, D6 vs. D21, and D6 vs. D28) for four different antigens, resulting in a total of 16 comparisons (4 antigens × 4 time points). The Bonferroni-adjusted *p*-value threshold for these comparisons was 0.00313 (0.05/16).

Additionally, IgG levels were compared between two experimental groups at D56 and D6 for four antigens. This analysis resulted in 8 independent comparisons (2 groups × 4 antigens). To correct for multiple comparisons, the Bonferroni-adjusted *p*-value threshold was set at 0.00625 (0.05/8).

For all statistical tests, *p*-values below the adjusted thresholds were considered statistically significant.

Calculations were performed using GraphPad Prism version 9 (GraphPad Software, San Diego, CA, USA).

## 3. Results

Primary CDI

a.Antibiotic-induced gut microbiota dysbiosis

We compared the gut microbiota in fecal samples on D6 before any antibiotic treatment, D3 after antibiotic cocktail administration, D1 before clindamycin administration, and D0 before challenge by *C. difficile*. The five predominant phyla observed in the initial microbiota, in order of abundance, were Bacteroidota, Bacillota (formerly Firmicutes), Verrucomicrobiota, Actinomycetota (formerly Actinobacteria), and Pseudomonadota (formerly Proteobacteria) (Figure 2A,B).

Following the antibiotic cocktail exposition from D6 to D3 before challenge, we observed a strong increase in the relative abundance of Pseudomonadota (2.9% to 19.5%) and Verrucomicrobiota (7.2% to 11.7%). This was accompanied by a strong decrease in Actinomycetota (5.9% to 0.001%) and Bacteroidota (61.9% to 40.9%). The profile was relatively stable between D3 and D1. The intraperitoneal administration of clindamycin on D-1 before challenge further increased the Pseudomonadota abundance compared to the antibiotics cocktail in drinking water alone, and reached an average proportion of around 50% at D0. Over the kinetics, the abundance of Bacillota was relatively stable (no significant variations). Alpha diversity was also greatly reduced (on average from 210 to 45 in observed richness between D6 and D0) after antibiotic treatment followed by clindamycin injection (Figure 2C) with a significant difference between D-6 and the other three times, regardless of the diversity index studied. Beta diversity analysis using the Bray–Curtis index indicated that the samples from D1 and D3 were close and well separated from D0 and D6 (Figure 2D and Appendix A). Finally, we conducted the linear discriminant analysis (LDA) effect size (LEfSe) algorithm, which was used to identify taxa specific to successive antibiotic treatments (Appendix A). Altogether, the microbiota signatures of the four timepoints are well defined.

b.Clinical monitoring

Animals were all symptomatic following the orogastric challenge by the 630Δ*erm*, and no symptoms were observed in animals challenged by the nontoxigenic strain. We observed an overall mortality of 15.7% that occurred between D2 and D7. The maximum mean weight loss was 13.1%, observed on D2, which was consistent with the clinical signs of CDI (Figure 3). Free toxins were detected in 100% of the animals on D2.

c.*C. difficile* dynamics of digestive colonization

As expected, none of the animals were colonized on D6. All mice were colonized by *C. difficile* on D2 (40/40) and were all still colonized at D7 (19/19). The digestive clearance of bacteria was observed from D14, as there were only 10/19 animals still colonized with *C. difficile* (52.6%), including six with direct detection on agar-based cultures (31.6%) and four in which *C. difficile* was only detected after an enrichment step (21%). At D21, only 1/13 mouse was still colonized with *C. difficile* (after enrichment step; 8%). Finally, none of the animals were still colonized at D28 (0/8) (Figure 4).

d.*Humoral* immune response

In this model, we described the kinetics of onset of the immune response following a primary CDI.

i.Primary immune response (IgM)

The IgM response was heterogeneous among the different groups (Figure 5).

Nevertheless, we observed a significant increase in anti-TcdA IgM level as soon as D7 (median = 0.075) in comparison to D6 (median = 0.004; *p* = 0.0016). Anti-TcdA IgM levels were also significantly higher at D14 (median = 0.206; *p* = 0.0007), and D21 (median = 0.161, *p* = 0.0016) in comparison to the D6 level. We also observed an increase in anti-TcdB IgM level as soon as D14 (median = 0.1245; *p* = 0.007) in comparison to D6 (median = 0.07). Anti-TcdB IgM levels were also significantly higher at D21 (median = 0.2690; *p* = 0.0016) in comparison to the D6 level.

Regarding the IgM response toward SlpA, we did not observe a significant increase in IgM at D7, D14, and D21 (medians = 0.054, 0.0675 and 0.064, respectively) compared to D-6 level (median = 0.042).

Finally, we observed a significant increase in anti-FliC IgM level as soon as D7 (median = 0.159; *p* = 0.0016), and at D14 and D21 (median = 0.1845; *p* = 0.0007; median = 0.234; *p* = 0.0016, respectively,) in comparison to D-6 (median = 0.077).

ii.Secondary immune response (IgG)

The IgG response was also heterogeneous among the different groups (Figure 6).

We have observed a later appearance of IgG directed against the toxins from D21 after infection. This increase was significant for anti-TcdA IgG at D21 and D28 (median = 0.103; *p* = 0.0008 and median = 0.4925; *p* < 0.0001, respectively) and for anti-TcdB IgG at D28 (median = 0.029; *p* = 0.0152) compared to the basal level detected in the D6 group (median = 0.0085 for TcdA and median = 0.0083 for TcdB). We observed a non-significant increase in anti-TcdB IgG at D21 in comparison to the D6 level (median = 0.0483; *p* = 0.0062).

In contrast to the toxins, no IgGs directed against surface antigens (SlpA and FliC) were detected in all animals on D7, D14, D21, and D28.

2.Boost impact at D28

We wanted to measure the impact on mice of a new exposure with *C. difficile* at D28 after the first challenge on the immune response.

a.Clinical monitoring

No animal was symptomatic or died following the orogastric challenge by the 630Δ*erm* or the nontoxigenic strain at D28. All animals were colonized with *C. difficile* after challenge. Free toxins were also detected in all animals of group G two days after reinfection. No free toxins were detected in animals of group H. After reinfection, no weight loss was observed among both groups, which was consistent with the absence of clinical signs of CDI.

b.*Humoral* immune response (IgG)

At D56, we observed a significant increase in anti-TcdA and anti-TcdB IgG in animals challenged by the 630Δ*erm* strain compared to the D6 level (median = 0.530; *p* < 0.0001 and median = 0.263; *p* < 0.0001 compared to median = 0.07, respectively) (Figure 7).

Mice challenged by the non-toxigenic strain (group H) did not produce any anti-TcdA or anti-TcdB IgG in comparison to the D6 level (median = 0.0745 and median = 0.0815 compared to median = 0.07, respectively). As for the first infection, no IgGs directed against surface antigens (SlpA and FliC) were detected in animals from both groups despite *C. difficile* colonization.

c.Mucosal immune response (IgA)

A heterogeneous anti-TcdA and anti-TcdB IgA mucosal response was observed at D56 in all animals from the group infected with the toxigenic strain 630Δ*erm* (Figure 8).

The levels of anti-TcdB IgA were lower (median = 0.1565) than the levels of anti-TcdA IgA (median = 0.46). As was observed for the serum anti-toxin IgG, we did not observe neither anti-TcdA nor anti-TcdB in animals challenged with the non-toxigenic strain. Regarding the surface protein response, unexpectedly, no difference between the levels at D6 and D56 in the two groups was observed for anti-SlpA and anti-FliC IgA.

## 4. Discussion

The main objective of this work was to describe and validate a model of primary mild-*C. difficile* infection (CDI) in a naïve host, which is intended to better reflect the evolution of CDI in humans.

Regarding animal models of CDI, the golden Syrian hamster or *Mesocricetus auratus* is preferably used as a lethal model of CDI and therefore does not reflect the natural course of CDI in humans [21]. Furthermore, with this model, it is impossible to monitor animals after CDI. Mouse models have long been considered as colonization models, but Chen et al. have shown that a mouse model (based on C57BL/6 mice) can be used as an infection model by varying strains and amounts of infecting inocula of *C. difficile* [23]. Thus, in these conditions, the course of infection could reflect mild CDI observed in humans, which spontaneously resolves. Moreover, this model is adapted to study the development of adaptive humoral and mucosal immune responses over time. While our findings provide valuable insights into the immune response to *Clostridioides difficile* infection in a murine model, it is important to acknowledge that these results may not fully translate to clinical practice without further validation in human studies.

We used Chen’s model modified by using the less virulent 630Δ*erm* strain. Gut microbiota dysbiosis analysis of the antibiotic cocktail used in this model was evaluated in order to explain CDI susceptibility. The microbiota analysis data are consistent with those described by Fachi et al. [34]. In fact, the administration of the antibiotic cocktail followed by an injection of clindamycin promotes a sharp increase in Pseudomonadota. Our data shows that the combination of the antibiotics cocktail and clindamycin injection is more effective than the antibiotics cocktail alone in impairing gut microbiota balance in our mouse model. Overall, alpha diversity also declines sharply, as expected. Interestingly, regarding beta diversity, we observe that the microbial communities are less modified between D3 (end of antibiotic cocktails in drinking water) and D1 (before clindamycin injection) in the first dysbiotic state, before reaching a second one at D0 (before clindamycin injection), which was even more distant from D6 than D3/D1.

Our decision to use vegetative cells is supported by previous study, specifically the study by Chen et al., who used vegetative forms in their mouse model of CDI [23]. The standard approach often involves the use of spores, due to their ability to survive in aerobic conditions and germinate under anaerobic conditions in the host [22]. However, several studies have demonstrated successful infection models using vegetative cells [35,36,37]. For instance, vegetative cells are known to produce the primary virulence factors, toxins A and B, responsible for disease pathology [38]. Regarding CDI evolution, our results indicate that mice challenged with vegetative forms exhibited significant disease symptoms combined with a significant weight loss, including a 15.7% mortality rate. The mean weight loss of surviving mice was 13.1% from basal levels with a nadir at D2, and a return to basal weight at D6. We also detected free toxins in stool from all mice two days after being challenged by the toxigenic strain. Batah et al. observed 50% mortality using the same infection protocol, but with the epidemic strain R20291 ribotype 027 [24]. These data confirm that the variability in clinical response to the infection depends on the strain used.

Using vegetative forms also led to a persistent colonization up to day 7 post challenge, and a colonization percentage approximately 50% by day 14 suggests that the disease was not solely due to pre-existing toxins in the inoculum but also involved active bacterial colonization and toxin production in the host. All mice were colonized with *C. difficile* until D14, but the colonization rates decreased from D7 and disappeared between D14 and D21 in almost all animals. We did not quantify *C. difficile* in colony-forming units (CFU) per mg of feces, but instead focused on the percentage of colonization. Our objective was to determine whether the animals were colonized and, if so, for how long. We also used an enrichment method with taurocholate to monitor bacterial clearance with low inocula. This approach met our primary goal based on our findings. This model can be further used to evaluate the factors influencing colonization. For this purpose, quantifying *C. difficile* in CFU per mg of feces would be beneficial. This quantification would provide more detailed insights into the bacterial load and its dynamics during the course of infection, thereby enhancing our understanding of colonization and clearance mechanisms.

Thus, the mice were indeed infected, and recovered from the infection without antibiotic treatment. These data are consistent with those of Chen et al. regarding the symptomatology of this model, but they did not observe any deaths in their evaluation. However, the authors do not mention euthanasia and do not describe ethical endpoints of experiment [23].

We described a relevant and appropriated model of CDI in mice to assess the establishment of an adaptive immune response directed against toxins and major colonization factors of *C. difficile* in the different groups of mice sampled at several endpoints. Protocol in endpoint sampling was mandatory in order to obtain a large volume of blood. Indeed, the development of antibody detection tests initially required a large volume of sera. Further studies should have a protocol with paired animals and blood sampling to keep the animal alive and within the weight limit.

Regarding primary immune responses, we observed an early IgM response directed against toxins, mostly TcdA, but also against FliC as early as seven days after infection. This response was heterogeneous between groups, but increased until D21 for all tested antigens. Due to limited serum availability and the need for test repeatability, we were unable to measure IgM levels at day 28 for all samples. However, we did measure IgM levels in two mice at this time point, and the results were not significantly different from those at day 21. A larger study with more mice would be valuable to specifically evaluate the decline of IgM over time.

Subsequently, we observed an increase in anti-TcdA and anti-TcdB IgG as early as D21 after infection. This response is specific, because animals colonized with the non-toxigenic strain did not produce anti-toxin antibodies.

However, unexpectedly, we did not observe any IgG response directed against SlpA or FliC. These results are in agreement with those of Johnston et al. [39]. In fact, using the same model of infection in C57BL/6 mice and reinfecting the surviving mice five weeks after recovery, they found the presence of IgG directed against TcdA and TcdB in sera of the mice before reinfection, but the absence of antibodies directed against the surface antigens of *C. difficile* (immunofluorescence and Western blot techniques).

The first hypothesis was that the C57Bl/6 mouse, possessing only the H2-IAb allele of major histocompatibility complex (MHC) class II, was unable to recognize SlpA or FliC [40]. We therefore performed a prediction of epitope binding by MHC class II of the C57Bl/6 mouse using the IEDB prediction tool [41]. The analysis showed that the C57Bl/6 mouse was able to recognize SlpA or FliC on more than three epitopes. Another explanation could be a lack of class switching. The response directed against SlpA and FliC in mice could correspond to an adaptive immune response of the thymo-independent type, characterized by an early appearance of IgM (as early as D7), a low rate of class switching and thus a low rate of IgG secretion during the primary response. The thymo-independent response involves BCR and TLR receptors located on the surface of B cells (e.g., LPS with TLR-4), particularly those in the marginal zone [42,43]. The interaction of SlpA and FliC with TLR-4 and TLR-5, respectively, supports this hypothesis [44,45]. The response directed against TcdA and TcdB is thought to be a thymo-dependent immune response, because an infection of major histocompatibility complex type II KO mice results in 100% death in this model [39].

Other hypotheses to explain the absence of an anti-SlpA or anti-FliC IgG-type response have been put forward: (1) the need for a second contact with the bacteria and/or (2) that the response directed against SlpA and FliC would simply be mucosal.

Including group G with a boost at D28 was mainly to determine if a second contact with *C. difficile* will enhance anti-SlpA and anti-FliC IgG responses or if we will find anti-SlpA and anti-FliC IgA.

Regarding the boost on D28, after recovering from a first episode of CDI, the anti-TcdA and anti-TcdB IgG response increased significantly on D56 compared to the levels observed on D6. This response was also heterogeneous within the groups. The anti-TcdB IgG levels observed were also increased, but were lower than those observed for anti-TcdA IgG. The same difference was observed in patients infected with *C. difficile* [6]. For anti-SlpA and anti-FliC IgG, a second contact with *C. difficile* failed to produce an immune response against these surface proteins, as no difference was observed for both groups G and H between D6 and D56.

A heterogeneous mucosal anti-TcdA and anti-TcdB IgA response was observed on D56 in all animals in the group infected with the toxigenic 630Δ*erm* strain. The anti-TcdB IgA levels were also lower than the anti-TcdA IgA levels, similar to what we observed for serum anti-toxin IgG.

Despite reinfection with the same toxigenic strain four weeks after the initial challenge, we did not observe an elevation of anti-SlpA or anti-FliC IgG. Another explanation for the lack of IgG response against SlpA and FliC could be the availability of antigens for dendritic cells. Regarding toxins, many studies have shown that they are secreted into the intestinal lumen and can diffuse through the mucus to intestinal and immune cells. But, what about surface proteins outside the direct interactions between the bacteria and these cells? After the ingestion of antibiotics, SlpA and FliC could be unmasked following bacterial lysis, making them available and accessible to epithelial and immune cells. This contact would then lead to an adaptive immune response with presentation to T cells and generation of memory IgG antibodies. In perspective, we could repeat this study by treating the mice with vancomycin after infection to evaluate the effect of this treatment on the synthesis of serum IgG and mucosal IgA specifically directed against SlpA and FliC.

Finally, we observed that animals after a boost at D28 were not symptomatic, despite a colonization with the detection of free toxins. This model can also be evaluated to study recurrent CDI and answer if anti-toxins IgG could prevent further infections. For this, specifics control groups should be used, alongside evaluating the neutralizing effect of produced antibodies, using µMT mice or an anti-CD20 antibody.

## 5. Conclusions

To the best of our knowledge, this is the first description of a C57Bl/6 mouse model infected by *Clostridioides difficile* strain 630Δ*erm* using vegetative cells and a specific antibiotic cocktail (kanamycin, gentamicin, colistin, metronidazole, vancomycin, followed by clindamycin). We developed a relevant model of primary mild-CDI in naive C57BL/6 mice, allowing for the study of the dynamic of the implementation of an immune adaptive response following CDI.

We detected serum IgM early, directed against *C. difficile* toxins A and B after CDI. Serum IgG anti-TcdA and TcdB rapidly appeared from D21 after a first episode, and significantly increased following a second contact with *C. difficile* following reinfection. Moreover, a high level of IgA, directed against both toxins, was also detected in the cecal contents. This model ought to be utilized to study human infection, and could enable detailed studies on immune responses and recurrence, microbiota dynamics, and the impact of antibiotics.

## Figures and Tables

**Figure 1 microorganisms-12-01933-f001:**
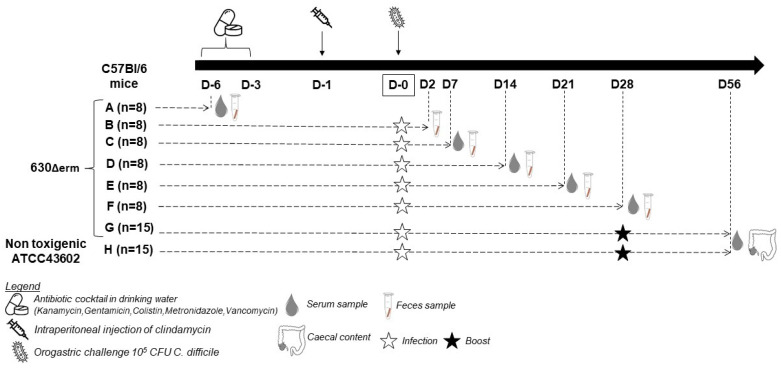
Experimental conditions and the protocol of infection, with a sample protocol. Mice from the groups G and H were reinfected at D28 following the same protocol used at D0, including antibiotic cocktails from D22 to D25 and the intraperitoneal infection of clindamycin at D27.

**Figure 2 microorganisms-12-01933-f002:**
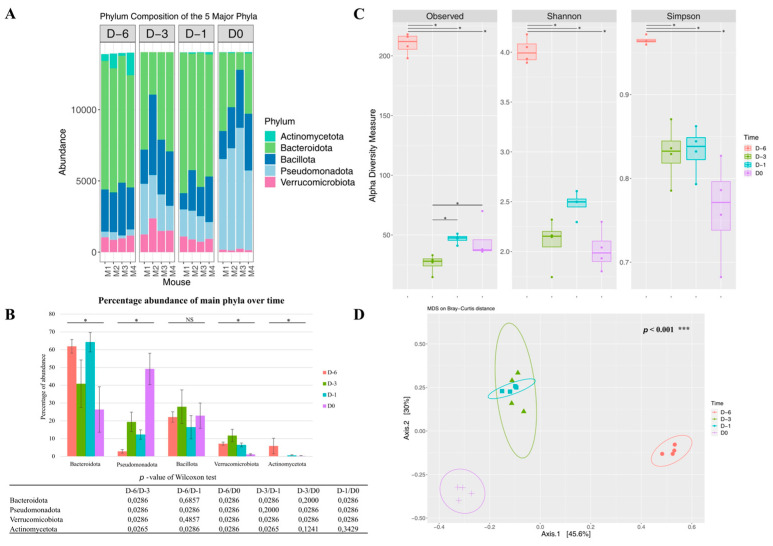
Microbiota changes after antibiotic treatment. (**A**)—Stacked bar-plot representation of fecal microbiota compositions before antibiotic treatment (D-6), after 3 days of the antibiotic cocktail (D-3), just before the clindamycin administration (D-1), and on the day of challenge (D0) with taxonomic features collapsed at the level of phyla. (**B**)—Percentage abundance of main phyla over time. *p* values less than 0.05, indicated by the stars above the graph, indicate that the difference between groups is significant, determined by Kruskal–Wallis tests. If it is not significant, it is indicated by NS. A Wilcoxon test was performed and the *p*-value is shown under the graph in the table. The smallest *p*-value you can obtain when comparing two samples of size 4 and 4 is 0.0286. (**C**)—Alpha diversity from fecal microbiota of mice at these same different time points. *p* values less than 0.05, marked with stars on the graph, indicate that the difference between groups is significant, determined by a Wilcoxon test. Non-significant 2-to-2 comparisons are not shown. (**D**)—MDS plot using Bray–Curtis dissimilarity based on phylum-level ASVs from fecal microbiota of mice at these same different time points. Statistics were calculated using PERMANOVA with 999 permutations, *** *p* < 0.001.

**Figure 3 microorganisms-12-01933-f003:**
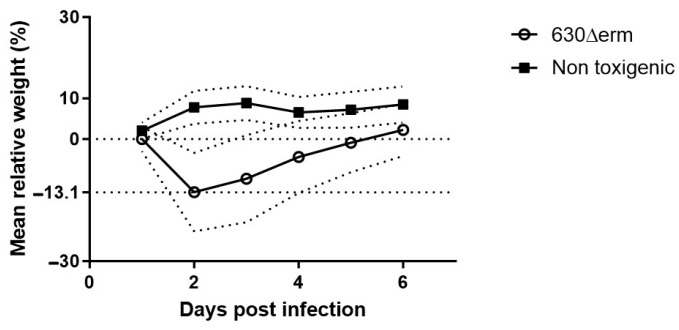
Mean weight loss relative to the initial weight measured on D0 after orogastric challenge. The dotted lines around the main line of weight loss represent the standard deviations.

**Figure 4 microorganisms-12-01933-f004:**
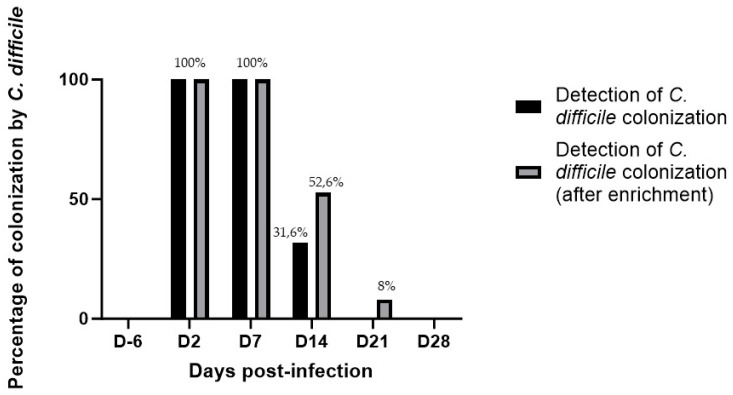
Colonization rate of infected mice after orogastric challenge by 10^5^ CFU of *C. difficile* for mice of groups A to F. The data presented are descriptive and illustrate the progression of *C. difficile* colonization over time without statistical comparison between time points.

**Figure 5 microorganisms-12-01933-f005:**
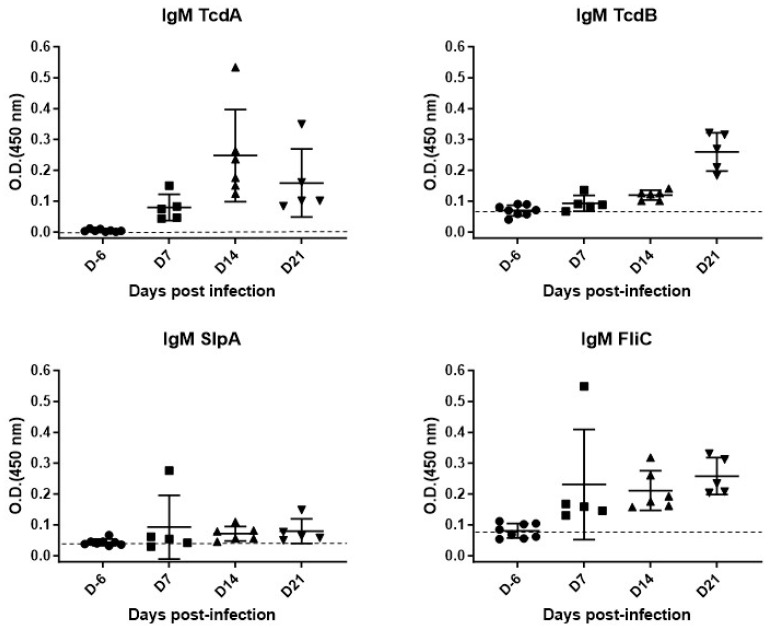
Kinetics of IgM appearance against TcdA, TcdB, SlpA, and FliC after a primary CDI in C57Bl/6 mice.

**Figure 6 microorganisms-12-01933-f006:**
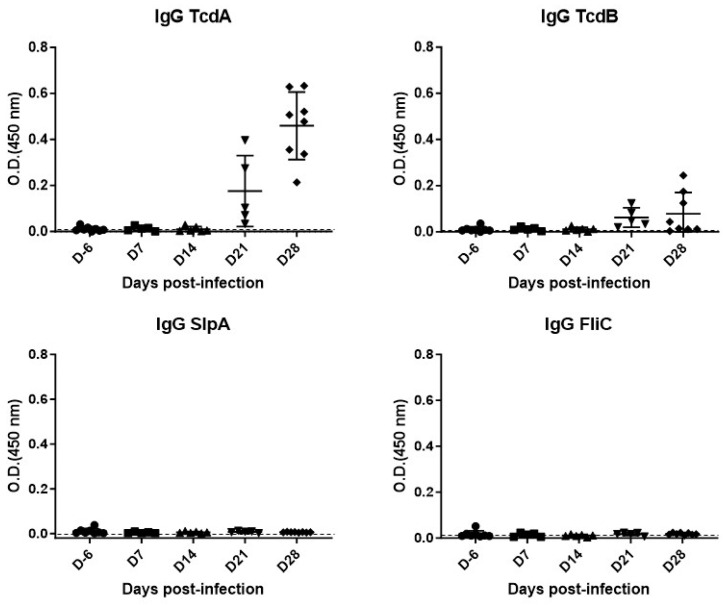
Kinetics of IgG appearance against TcdA, TcdB, SlpA and FliC after a primary CDI in C57Bl/6 mice.

**Figure 7 microorganisms-12-01933-f007:**
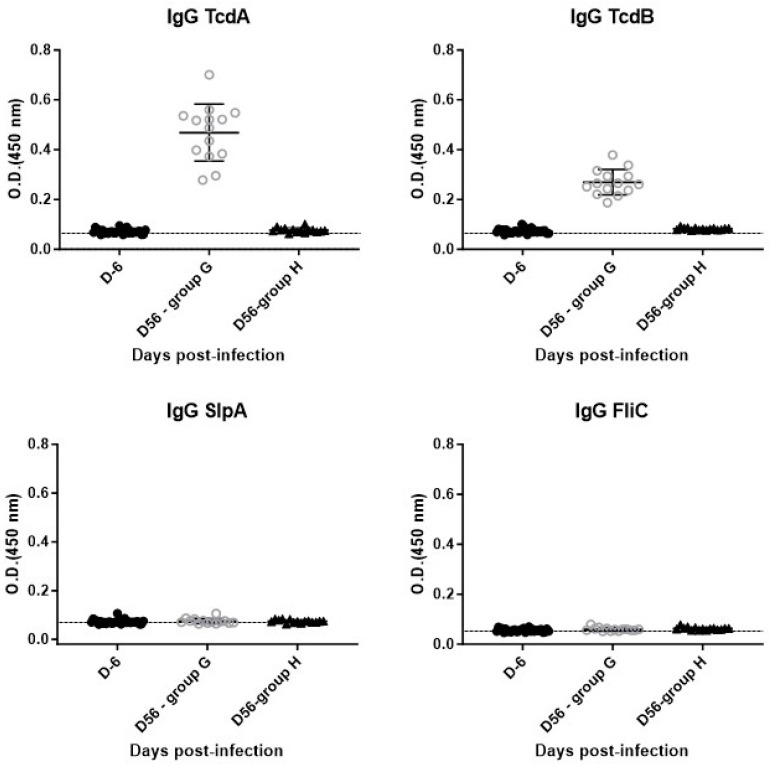
Level of IgG against TcdA, TcdB, SlpA and FliC at D56 in animals from group G (630Δ*erm*) and group H (Non-toxigenic).

**Figure 8 microorganisms-12-01933-f008:**
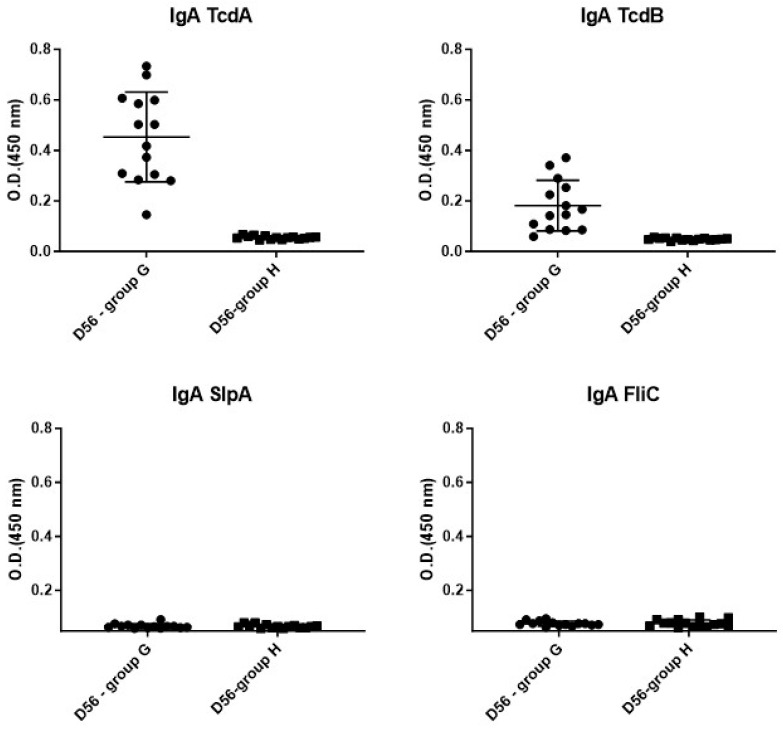
Level of IgA against TcdA, TcdB, SlpA and FliC at D56 in animals from group G (630Δ*erm*) and group H (Non-toxigenic).

## Data Availability

The data presented in this study are available on request from the corresponding author due to privacy restrictions.

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
