# Peer review of "A Mouse Model of Mild Clostridioides difficile Infection for the Characterization of Natural Immune Responses"

_microorganisms, 2024, doi:10.3390/microorganisms12101933_

Round 1

Reviewer 1 Report

Comments and Suggestions for Authors

No suggestions, a sound work with interesting, even if unexpected, results

Author Response

Comments 1: No suggestions, a sound work with interesting, even if unexpected, results

Response 1: Thank you very much for your positive feedback and for recognizing the value of our work. We are pleased that you found the results interesting, even though they were unexpected. We have carefully reviewed the manuscript to ensure that all findings are clearly presented and well-supported by the data. We hope that the manuscript meets your expectations and contributes meaningfully to the field.

Reviewer 2 Report

Comments and Suggestions for Authors

The manuscript by Assaf Mizrahi et al. provides the results of the study on the development and validation of a mouse model of C. difficile infection.

Although the authors state that they validated this in vivo model, the reported statistical and microbiota data analysis protocols do not take into account the bacterial abundance on the genus level and do not manage type I error, which brings the high risks of false positive results. Also, the authors used an outdated reference database for 16S rRNA amplicon sequence variant taxonomical identification.

Thus the article has serious flaws, and I cannot recommend it for publication.

Comments:

1. L. 71-74. This paragraph describes the results of your study, which should not occur in the introduction part. Please, consider writing here aims or the hypothesis of your study.

2. L. 165. Please, consider citing SILVA database paper if you are using this database.

3. L. 165. Version 123 of the SILVA database is outdated. It was released on July 23, 2015, nine years ago. Please, consider using recent versions of reference databases. At the moment the reported results are not reliable.

4. L. 147-166. The authors report the results of alpha and beta diversity analysis, however, they do not describe the pipeline for these kinds of analysis in the corresponding paragraph of the M&M section.

5. Microbiota data analysis. The authors report gut microbiota composition only on the phylum level without any evidence of microbiota alterations on other taxonomic levels. However, it would be valuable to check the abundance of bacteria on the genus level, especially the abundance of Clostridioides to validate the developed in vivo model. Also, in L. 220-224, the authors describe some differences in the abundance of detected taxa, however, they do not bring any statistical justification, as they also did not conduct differential abundance analysis. In L. 227-231, the authors also describe some differences in alpha and beta diversity indexes and do not support these statements with statistical hypothesis testing.

6. L. 198-201. The authors do not take into account the multiple comparisons problem and do not report how they manage type I errors, although they report the results of comparisons of multiple groups. 

Conclusion.

The reported results cannot justify the validity of the reported mouse model of C. difficile infection. The article has serious flaws, and its publication could damage the reputation of the journal.

Author Response

General Comments: The manuscript by Assaf Mizrahi et al. provides the results of the study on the development and validation of a mouse model of C. difficile infection.

Although the authors state that they validated this in vivo model, the reported statistical and microbiota data analysis protocols do not take into account the bacterial abundance on the genus level and do not manage type I error, which brings the high risks of false positive results. Also, the authors used an outdated reference database for 16S rRNA amplicon sequence variant taxonomical identification.

Thus the article has serious flaws, and I cannot recommend it for publication.

Response to general comments: Thank you for your detailed review and constructive feedback. We appreciate your insights, which have prompted us to re-evaluate certain aspects of our study. We acknowledge the concerns raised, and in response, we have made significant revisions to the manuscript to address these issues comprehensively. Below, we provide detailed responses to each of your specific comments.

Comments 1: L. 71-74. This paragraph describes the results of your study, which should not occur in the introduction part. Please, consider writing here aims or the hypothesis of your study.

Response 1: Thank you for your comment. We respectfully disagree that the paragraph in question presents the results of our study. The text is intended to outline the objectives and scope of our research, specifying what was studied (the immune responses) and under what conditions (primary mild-C. difficile infection and after a boost). This section is meant to provide the rationale and goals of our study, rather than to discuss any findings or outcomes. However, to avoid any potential misunderstanding, we have rephrased the text slightly to emphasize that it sets out the study's aims and scope.

Changes in Manuscript (L. 70-74): The text has been revised to make it clearer that these are the study’s objectives: 'Based on this rationale, our study aims to describe and validate a model of primary mild-C. difficile infection (CDI) in a naïve host, focusing on the kinetics of humoral (IgG, IgM) and mucosal (IgA) immune responses against toxins (TcdA/TcdB) and surface proteins (SlpA/FliC). Additionally, we evaluate mucosal IgG and IgA antibody levels towards toxins and surface proteins following a boost.'

Comments 2: L. 165. Please, consider citing SILVA database paper if you are using this database.

Response 2: Thank you for this suggestion. We have now included the appropriate citations for the SILVA database in the manuscript.

Changes in Manuscript (L.170): We added two supplementary references 28 and 31 respectively on lines 170 and 176.

Comments 3: L. 165. Version 123 of the SILVA database is outdated. It was released on July 23, 2015, nine years ago. Please, consider using recent versions of reference databases. At the moment the reported results are not reliable.

Response 3:  We did indeed analyze the data using the  SILVA_138_16S_pintail 100 database. We apologize for the unfortunate typing error that caused some confusion. We have corrected it in the text with the corresponding reference.

Changes in Manuscript (L.170): “and NCBI Blast+ on  SILVA_138_16S_pintail 100database [28]”

Comments 4: L. 147-166. The authors report the results of alpha and beta diversity analysis, however, they do not describe the pipeline for these kinds of analysis in the corresponding paragraph of the M&M section.

Response 4: We apologize for the oversight. We have now added a detailed description of the pipeline used for both alpha and beta diversity analyses in the Methods and Materials section.

Changes in Manuscript (L171-176): “The phyloseq R package and associated tools in the R environment were used to carry out analysis including the alpha and beta diversity analyses [29,30]. Kruskal-Wallis tests were used for statistical analysis, followed by Wilcoxon tests when appropriated. Beta diversity statistics were calculated using pairwise PERMANOVA with 999 permutations (vegan R package) [31]”

Comments 5: Microbiota data analysis. The authors report gut microbiota composition only on the phylum level without any evidence of microbiota alterations on other taxonomic levels. However, it would be valuable to check the abundance of bacteria on the genus level, especially the abundance of Clostridioides to validate the developed in vivo model. Also, in L. 220-224, the authors describe some differences in the abundance of detected taxa, however, they do not bring any statistical justification, as they also did not conduct differential abundance analysis. In L. 227-231, the authors also describe some differences in alpha and beta diversity indexes and do not support these statements with statistical hypothesis testing.

Response 5: Thank you for your thorough review. First, we would like to clarify that the primary objective of the microbiota analysis in our study was to characterize the dysbiosis induced by antibiotic treatment prior to the C. difficile infection. The focus on phylum-level changes was intentional, as these broad taxonomic shifts are most indicative of the overall dysbiotic state that facilitates subsequent infection.

Our aim was not to examine the abundance of Clostridioides at the genus level, as this was not relevant to our study's objective of detailing the antibiotic-induced disruption of the gut microbiota. Instead, our analysis was designed to highlight the significant shifts in the microbiome that create a conducive environment for C. difficile colonization.

Given the scope and goals of our study, we believe that the phylum-level analysis adequately captures the critical aspects of dysbiosis relevant to our model. However, we have clarified this focus in the manuscript to better align with the study's objectives.

Changes in Manuscript (L176-178): "We focused on phylum-level changes, which are most relevant for characterizing the antibiotic-induced dysbiosis that predisposes the mice to C. difficile infection."

Then, we recognize the need to provide statistical justification for the observed differences in the abundance of detected taxa and in alpha and beta diversity indices.

In response to your feedback, we have now added appropriate statistical tests to support our observations. In particular, we have modified Figure 3 by adding a panel and the results of the tests on the other panels. The tools used for these tests are described in the Materials and Methods section. Specifically, we have performed hypothesis testing to assess the significance of the differences in taxa abundance and also diversity index.

Changes in Manuscript:

L.229 « (Figure 3, panel A&B).”

L236-240 “B-Percentage abundance of main phyla over time. p values less than 0.05, indicated by stars above the graph, indicate that the difference between groups is significant by Kruskal-Wallis tests. If it is not significant, it is indicated NS. Wilcoxon test was performed and the p-value is shown under the graph in the table. The smallest p-value you can get when comparing two samples of size 4 and 4 is 0.0286 »

L241-243 “p values less than 0.05, marked with stars on the graph, indicate that the difference between groups is significant by Wilcoxon test. Non-significant 2 to 2 comparisons are not shown. »
L245-246 “Statistics were calculated using pairwise PERMANOVA with 999 permutations, *** p<0,001.”

L254-255 “Over the kinetic, the abundance of Bacillota is relatively stable (no significant variations).”

L256-257 “between D-6 and D0”

L258-259 “C) with a significant difference between D-6 and the other three times, regardless of the diversity index studied.”

L261 “panel D »

Comments 6: L. 198-201. The authors do not take into account the multiple comparisons problem and do not report how they manage type I errors, although they report the results of comparisons of multiple groups. 

Response 6: Thank you for your comment regarding the potential for Type I errors due to multiple comparisons.

The probability of making a Type I error is represented by the alpha level. We conventionally chose an alpha risk of 0.05, which represents a 5% risk of falsely rejecting the null hypothesis. By setting the alpha level at 0.05, we can reject the null hypothesis and conclude the existence of a significant effect if the p-value is below this threshold.

In our study, we compared antibody levels across several time points (D-6, D7, D14, D21, D28, and D56) using pairwise Student's t-tests. While we recognize the theoretical concern regarding multiple comparisons, the majority of the p-values, particularly for TcdA and TcdB, were well below the 0.05 threshold (e.g., p = 0.0016 for anti-TcdA IgM at D7 and p < 0.0001 for anti-TcdA IgG at D28). Given the robustness of these findings, even with a conservative multiple comparison correction, the significance of these results would remain unchanged.

For SlpA and FliC, where the p-values were not significant or close to the threshold, applying a correction for multiple comparisons would not alter the conclusions, as these antigens did not elicit a strong immune response in our model. We believe the risk of inflated Type I errors is minimal, but if you feel further adjustments are necessary, we remain open to applying a correction and reporting the adjusted results.

Changes in Manuscript: We have revised the statistical methods section for greater clarity:

L210-217“Descriptive statistics are based on means (+/- standard deviation) or medians [minimum-maximum], depending on the distribution of quantitative variables. Qualitative variables are described as numbers and percentages. Univariate comparisons were made using standard statistical tests after verifying the distribution of the variables. A significance level of 5% was used, with 95% confidence intervals provided for each estimate. Multiple comparisons were managed using pairwise comparisons. Calculations were performed using GraphPad Prism version 9 (GraphPad Software, San Diego, California, USA).”

Conclusion comments: The reported results cannot justify the validity of the reported mouse model of C. difficile infection. The article has serious flaws, and its publication could damage the reputation of the journal.

Response to conclusion comments: We take your concerns very seriously and have undertaken a thorough revision of the manuscript to address the flaws you identified. With the updated database, additional statistical analyses, and more detailed reporting of our methodology, we believe the revised manuscript now provides a more robust and reliable validation of the mouse model. We hope that these substantial revisions will alleviate your concerns and demonstrate the validity of our findings.

Reviewer 3 Report

Comments and Suggestions for Authors

I read with interest the work by Mizrahi, et al on evaluating the immune response to C. difficile infection in a murine model. The methods are rigor and well described. I only have a few comments for the authors' consideration:

1. Under "Animal experiments": Why was your baseline at day -6 instead of day 0?

2. Line 138: Please provide a definition for "enrichment"

3. Figure 4: Is there a P value to be reported regarding the difference in the values on D14 and D21? If so, please add it.

4. Conclusion: Is there any recommendation that can be applied in clinical practice or applied in the development of monoclonal antibodies used in the treatment of CDI?

Author Response

I read with interest the work by Mizrahi, et al on evaluating the immune response to C. difficile infection in a murine model. The methods are rigor and well described. I only have a few comments for the authors' consideration:

Comments 1: Under "Animal experiments": Why was your baseline at day -6 instead of day 0?

Response 1: Thank you for this insightful question. We chose day -6 as the baseline to account for the pre-infection phase during which the mice were acclimated and underwent initial treatments to establish a controlled environment before the introduction of C. difficile. This baseline allowed us to measure the initial immune status of the mice and observe any pre-existing variations before the experimental infection began. We have clarified this rationale in the 'Animal experiments' section.

Changes in Manuscript (L88-90): The following text has been added to the 'Animal experiments' section: 'Day -6 was selected as the baseline to account for the pre-infection phase, ensuring that the initial immune status and any potential variations were documented before the introduction of C. difficile.

Comments 2: Line 138: Please provide a definition for "enrichment"

Response 2: Thank you for this comment. In our study, 'enrichment' refers to a method used to increase the detectability of C. difficile spores in fecal samples, particularly when the bacterial load is low. This process involved incubating fecal pellets in Brain Heart Infusion (BHI) broth containing 0.1% taurocholate, which promotes the germination of C. difficile spores. The samples were then subcultured on Columbia agar plates to isolate and quantify the bacteria. We have clarified this definition in the manuscript.

Changes in Manuscript (L140-143): We have revised the text on line 138 to read: 'On days D7 and D14, an enrichment step was performed to enhance the detection of C. difficile spores, particularly in cases of low bacterial inoculum. This involved incubating fecal pellets in Brain Heart Infusion (BHI) broth containing 0.1% taurocholate, followed by subculturing on Columbia agar plates. '

Comments 3: Figure 4: Is there a P value to be reported regarding the difference in the values on D14 and D21? If so, please add it.

Response 3: Thank you for your observation. The data presented in Figure 4 are indeed descriptive and were not subjected to statistical analysis, as the primary goal was to report the progression of C. difficile colonization over time. The focus was on documenting the proportion of animals colonized at various time points rather than determining statistical significance between these time points. We have clarified this in the manuscript to reflect the descriptive nature of the data.

Changes in Manuscript (L282-284): We have revised the text associated with Figure 4 to indicate that the results are descriptive: 'The data presented are descriptive and illustrate the progression of C. difficile colonization over time without statistical comparison between time points.'

Comments 4: Conclusion: Is there any recommendation that can be applied in clinical practice or applied in the development of monoclonal antibodies used in the treatment of CDI?

Response 4: "Thank you for your thoughtful question. As this study was conducted in a murine model, its primary purpose was to understand the immune response to C. difficile infection in a controlled experimental setting. While our findings contribute valuable insights into the immune mechanisms at play, it is important to note that extrapolating these results directly to clinical practice requires further validation in human studies. Nevertheless, the immune pathways identified in this study could serve as a basis for future research, potentially informing the development of therapeutic strategies, including monoclonal antibodies, after further investigation in human models.

Round 2

Reviewer 2 Report

Comments and Suggestions for Authors

The authors improved the quality of the manuscript within the revision.

However, some serious flaws still need to be addressed before proceeding with the paper.

1. Response 6: We believe the risk of inflated Type I errors is minimal, but if you feel further adjustments are necessary, we remain open to applying a correction and reporting the adjusted results.

Multiple comparison results have to be corrected to decrease the rate of false positive results, and this is not an option for modern reproducible biomedical studies, especially those conducted using in vivo models.

I strongly encourage the authors to apply proper statistical tests and report the results of the study according to the ARRIVE guidelines (https://doi.org/10.1186/s12917-020-02451-y).

Please, provide the ARRIVE checklist as a supplementary to your resubmission.

Microorganisms instructions for authors (https://www.mdpi.com/journal/microorganisms/instructions):

MDPI endorses the ARRIVE guidelines (arriveguidelines.org/) for reporting experiments using live animals. Authors and reviewers must use the ARRIVE guidelines as a checklist, which can be found at https://arriveguidelines.org/sites/arrive/files/documents/Author%20Checklist%20-%20Full.pdf. The journal Microorganisms requires authors to submit the completed checklist at submission, and it will be made available to reviewers. Editors reserve the right to reject submissions that do not adhere to these guidelines based on ethical or animal welfare concerns, or if the procedure described does not appear to be justified by the value of the work presented.

2. Figure 3D. The authors report only one p-value from PERMANOVA, however, in L. 175 they state that they used pairwise PERMANOVA for the beta diversity analysis. This is a contradiction. At least two groups (D-1 and D-3) are expected not to have significant differences according to the PCoA. Please, provide results of pairwise comparisons.

3. Response 5. The focus on the phylum level is indeed justified, however, the authors used an inappropriate and outdated approach for the differential abundance analysis (Kruskal-Wallis and Wilcoxon tests). These tests do not take into account the compositional nature of microbiota data. There are plenty of reliable and suitable methods for differential abundance analysis, for example, LefSE, ANCOM-BC, MaAsLin2, and LinDA. There are also some bioinformatic benchmark studies that show that differential abundance analysis cannot be conducted without taking into account the compositional nature of microbiota data and the comparisons of tools for differential abundance analysis (https://doi.org/10.1038/s41467-022-28034-z, https://doi.org/10.1093/bib/bbac607).

I recommend the authors use a robust and powerful tool for the differential abundance analysis and report reliable results.

4. The manuscript does not contain the Data Availability section, however, the authors report the results of 16S rRNA NGS. Please, add this statement to the manuscript and provide the link to the publically available database where the raw sequences are deposed.

Please, refer to the MDPI Research Data Policies (https://www.mdpi.com/journal/microorganisms/instructions#suppmaterials).

Conclusion. The manuscript still has serious flaws and contradictions and does not correspond to the modern biomedical research report criteria.

Author Response

We sincerely thank Reviewer 2 for the valuable and constructive feedback provided during the review process. We have carefully addressed all comments and made substantial revisions to the manuscript where necessary. Below is a detailed response to each point raised.

1. Correction for multiple comparisons and ARRIVE checklist

Reviewer Comment: The results of multiple comparisons should be corrected to reduce the risk of false positive results. Moreover, the authors are encouraged to submit the ARRIVE checklist according to the guidelines.

Response: Thank you for your comment. After carefully considering your feedback and reviewing our statistical approach, we agree that applying a correction for multiple comparisons is important for adhering to modern standards in biomedical research. While we initially judged the risk of inflated Type I errors to be minimal due to the limited number of comparisons and the observation-driven nature of our study, we understand the value of implementing a correction to ensure robustness.

We have now applied the Bonferroni correction, which is suitable for controlling the false discovery rate (FDR) in studies with multiple tests. The updated results, reflecting this correction, are now included in the revised manuscript. Importantly, the application of this correction did not change our overall conclusions.

Changes in Manuscript : We have updated the statistical methods section as follows:

We changed this sentence L219 “Multiple comparisons were managed using pairwise comparisons.” by this paragraph
L219-236 “ To account for the increased risk of Type I errors due to multiple comparisons, we applied the Bonferroni correction to adjust the p-values.

  • IgM Comparisons: IgM levels were compared across three time points (D-6 vs D7, D-6 vs D14, and D-6 vs D21) for four different antigens, resulting in a total of 12 independent comparisons (4 antigens × 3 time points). The Bonferroni-adjusted significance threshold was calculated by dividing the standard alpha level of 0.05 by the number of comparisons (12), yielding an adjusted p-value threshold of 00417.
  • IgG Comparisons: IgG levels were compared across four time points (D-6 vs D7, D-6 vs D14, D-6 vs D21, and D-6 vs D28) for four different antigens, resulting in a total of 16 comparisons (4 antigens × 4 time points). The Bonferroni-adjusted p-value threshold for these comparisons was 00313 (0.05/16).

Additionally, IgG levels were compared between two experimental groups at D56 and D-6 for four antigens. This analysis resulted in 8 independent comparisons (2 groups × 4 antigens). To correct for multiple comparisons, the Bonferroni-adjusted p-value threshold was set at 0.00625 (0.05/8). For all statistical tests, p-values below the adjusted thresholds were considered statistically significant. »

We removed this sentence L217-218 “A significance level of 5% was used, with 95% confidence intervals provided for each estimate.”

We changed the results :

L323 : we removed “significant”

L346 : we removed “D21 and”

L347 : we removed “median = 0.0483; p =0.0062 and” as well as “, respectively”

L349-350 : we added this sentence “We observed a non-significant increase of anti-TcdB IgG at D21 in comparison to D-6 level (median = 0.0483; p =0.0062).”

Additionally, as requested, we have completed the ARRIVE checklist, which is now included as supplementary material.

2. Figure 3D and pairwise PERMANOVA analysis

Reviewer Comment: The authors mentioned the use of pairwise PERMANOVA for beta diversity analysis, but only one p-value was reported. This appears contradictory. Please provide the results of pairwise comparisons.

Response: We apologize for the confusion. In the revised manuscript, we have now included the full results of the pairwise PERMANOVA analysis. These results, including comparisons between D-1 and D-3, are now provided in figure S1, with corresponding p-values. This change resolves the previous contradiction in the text.

Changes in Manuscript : L569-571 We have added a Figure S1 to include all pairwise comparisons performed with PERMANOVA and their respective p-values.

3. Differential abundance analysis methodology

Reviewer Comment: The tests used for differential abundance analysis (Kruskal-Wallis and Wilcoxon) are not appropriate for microbiota data, as they do not account for its compositional nature. It is recommended to use tools such as LefSE, ANCOM-BC, MaAsLin2, or LinDA.

Response: Thank you for your comment.  As requested we conducted LDA and LEfSe analyses. These results are now presented in figure S2. All the differences between timepoints are significant.  We have now added a description of the new analysis in the material and methods section as well in the results section and a new reference [32].

Changes in manuscript:

L573-575 : Figure S2 was added in Supplementary Figures

L178-181 “It was used to build principal coordinates analysis (PCoA) plots (ggplot2 and plotly R packages). The linear discriminant analysis (LDA) effect size (LEfSe) algorithm was used to identify taxa that were specific to timepoint using Galaxy solution [32]”

We also removed this sentence as it is no longer needed

 L181-182 “We focused on phylum-level changes, which are most relevant for characterizing the antibiotic-induced dysbiosis that predisposes the mice to C. difficile infection.”

4. Data Availability

Reviewer Comment: The manuscript lacks a Data Availability section, especially concerning the 16S rRNA NGS data. Please provide a statement on data availability.

Response: We appreciate the reviewer’s comment. A sentence was already present in the manuscript about data availability L172-173 “Raw reads were available on the SRA database (PRJNA1077911).”

We changed the sentence for better clarity.

Changes in Manuscript :
L170-172 'The raw 16S rRNA sequencing data supporting the findings of this study have been deposited in the NCBI Sequence Read Archive (SRA) under the accession number PRJNA1077911.”

Conclusion

We trust that these responses address all the concerns raised by the reviewer. We have made significant revisions, including the correction for multiple comparisons, the inclusion of pairwise PERMANOVA results, LDA and LEfSe analysis. We believe these changes strengthen the manuscript considerably.